# Eosinophils Play a Surprising Leading Role in Recurrent Urticaria in Horses

**DOI:** 10.3390/vaccines12060562

**Published:** 2024-05-21

**Authors:** Katharina Birkmann, Fadi Jebbawi, Nina Waldern, Sophie Hug, Victoria Inversini, Giulia Keller, Anja Holm, Paula Grest, Fabia Canonica, Peter Schmid-Grendelmeier, Antonia Fettelschoss-Gabriel

**Affiliations:** 1Evax AG, Im Binz 3, 8357 Guntershausen, Switzerland; katharina@evax.ch (K.B.); fadi.jebbawi@usz.ch (F.J.); nina@evax.ch (N.W.); fabia.canonica@usz.ch (F.C.); 2Equine Department, Veterinary Faculty, Ludwig Maximilians University Munich LMU, Sonnenstrasse 14, 85764 Oberschleißheim, Germany; 3Department of Dermatology, University Hospital Zurich, Wagistrasse 18, 8952 Schlieren, Switzerland; peter.schmid@usz.ch; 4Faculty of Medicine, University of Zurich, 8032 Zurich, Switzerland; 5Central VetPharma Consultancy, Hauchsvej 7, 4180 Sorø, Denmark; 6Vetsuisse Faculty, Institute of Veterinary Pathology, University of Zurich, Winterthurerstrasse 268, 8057 Zurich, Switzerland; grest@vetpath.uzh.ch; 7Allergy Unit, Department of Dermatology, University Hospital Zurich, The Circle 59, 8058 Zurich-Airport, Switzerland

**Keywords:** urticaria, horses, IL-5 vaccination, eosinophils

## Abstract

Urticaria, independent of or associated with allergies, is commonly seen in horses and often shows a high reoccurrence rate. Managing these horses is discouraging, and efficient treatment options are lacking. Due to an incidental finding in a study on horses affected by insect bite hypersensitivity using the eosinophil-targeting eIL-5-CuMV-TT vaccine, we observed the prevention of reoccurring seasonal urticaria in four subsequent years with re-vaccination. In an exploratory case series of horses affected with non-seasonal urticaria, we aimed to investigate the role of eosinophils in urticaria. Skin punch biopsies for histology and qPCR of eosinophil associated genes were performed. Further, two severe, non-seasonal, recurrent urticaria-affected horses were vaccinated using eIL-5-CuMV-TT, and urticaria flare-up was followed up with re-vaccination for several years. Eotaxin-2, eotaxin-3, IL-5, CCR5, and CXCL10 showed high sensitivity and specificity for urticarial lesions, while eosinophils were present in 50% of histological tissue sections. The eIL-5-CuMV-TT vaccine reduced eosinophil counts in blood, cleared clinical signs of urticaria, and even prevented new episodes of urticaria in horses with non-seasonal recurrent urticaria. This indicates that eosinophils play a leading role in urticaria in horses, and targeting eosinophils offers an attractive new treatment option, replacing the use of corticosteroids.

## 1. Introduction

Urticaria, also called hives, is common in horses [1]. Comparably to humans, clinical signs of urticaria in horses are well-defined raised areas with lumps, wheals, or rings, which occur in the superficial dermis [2]. In severe cases, whole areas, e.g., the head or other mucosal areas, may become swollen. The pathogenesis of urticaria in horses primarily comprises IgE-mediated reactions, including mast cell degranulation accompanied by basophil degranulation, similar to type I allergies. Chemical mediators, such as vasoactive histamine, heparin, cytokines, prostaglandins, leukotrienes, and others, lead to an increase in vascular permeability (angioedema) and inflammation, which in turn causes wheal formation [2,3,4]. Skin biopsies reveal mild to moderate perivascular to interstitial aseptic dermatitis with numerous eosinophils and lymphocytes, including variable dermal oedema [4]. 

Comparably to humans, causes of urticaria in horses are various and include immunological causes and non-immunological causes, and urticaria may present with or without pruritus [2]. Immunological causes are, in particular, atopic dermatitis, food allergies, inhaled allergens, insect-bite hypersensitivity (IBH), vaccines, and drugs (penicillin, tetracycline, sulfonamides, neomycin, ciprofloxacin, streptomycin, aspirin, phenylbutazone, flunixin, phenothiazines, guaifenesin, ivermectin, moxidectin, pethidine, iron, dextrans, hormones, vitamin B complex, and liver extracts), vasculitis, contact with a substance or material, infections (bacterial (e.g., strangles), viral (e.g., horse-pox), fungal, parasitic (e.g., Trypanosoma equiperdum), protozoal), and snakebites. Non-immunologic causes include dermatographism and pressure, cold temperature, heat, sunlight, psychological stress, and exercise [2]. 

A challenge for the long-term clinical management and cure of equine recurrent urticaria is the identification of the underlying cause. Identification of the underlying cause and etiology of equine recurrent urticaria is often discouraging because recurrence is common [1]. Acute signs are often treated with systemic steroids, although severe side effects might occur, especially with long-term use, such as osteoporosis and laminitis [5]. Thus, there is a need for prevention and treatment options in horses which are affected by urticaria, and in particular, by recurrent urticaria. 

Here, we investigate the potential role of eosinophils in urticaria through histology, the gene expression profile of urticaria wheals, and by targeting eosinophils using the eIL-5-CuMV-TT vaccine in a case series of urticaria-affected horses. The eIL-5-CuMV-TT vaccine is a virus-like particle (VLP)-based vaccine, conjugated to multiple copies of equine Interleukin (IL)-5 to overcome B-cell tolerance, inducing anti-IL-5 antibodies and reducing levels of equine IL-5, the key cytokine exerting a central role in the differentiation, recruitment, survival, and degranulation of eosinophils [6,7]. Our data reveal that eosinophils were indeed found to infiltrate the skin during urticaria flare-ups, accompanied by a significant upregulation of chemoattractant genes to recruit eosinophils. Furthermore, the eIL-5-CuMV-TT vaccine was able to treat and prevent urticaria flare-ups in vaccinated horses.

## 2. Materials & Methods

### 2.1. Horses

All of the horses involved in this study were client-owned horses. All clinical studies had been approved by the respective cantonal veterinary authorities, license numbers 25152, 28780, 29968, 33558, and 33608. All horse owners gave written informed consent. All horses were located in Switzerland. *IBH & (seasonal) urticaria with eIL-5-CuMV-TT vaccination.* Three Icelandic horses from an IBH trial [6,7,8] showed additional seasonal clinical signs of urticaria. E-UAS was followed during an untreated year, a placebo year, and up to four vaccination years. *Recurrent urticaria-affected (non-IBH) & healthy horses for biopsy collection:* For 2 mm punch biopsies, including 12 recurrent urticaria-affected (no IBH) non-Icelandic horses and 24 healthy non-Icelandic horses. For 6 mm punch biopsies, including 6 recurrent urticaria-affected (no IBH) non-Icelandic horses. *Case reports recurrent urticaria (non-IBH) with eIL-5-CuMV-TT vaccination:* two non-Icelandic horses with recurrent urticaria (no IBH) received vaccination.

### 2.2. Exploratory Urticaria Activity Score (E-UAS)

An Urticaria Activity Score (UAS) was applied, which examines the severity of the urticaria using the area of hives on the affected skin of the horse. An exploratory urticaria activity (E-UAS) score ranges from 0 to 3, wherein 0 corresponds to no urticaria (hives/wheels), 1 corresponds to up to one-third (≤1/3) of the body affected, 2 corresponds to more than one-third up to half of the body affected (1/3 < x ≤ 1/2), and 3 corresponds to more than half of the whole body affected (>1/2). A similar test and score for the determination of urticaria symptoms has been established for human urticaria [9].

### 2.3. Blood Collection and Differential Blood Analysis

Blood was collected from the V. jugularis at the intersection of the proximal to median third of the neck. For the differential blood analysis, 3 mL of fresh EDTA blood was collected in tubes provided by IDEXX Diavet (Freienbach, Switzerland) and was measured by IDEXX Diavet.

### 2.4. Skin Punch Biopsies

Skin punch biopsies (Stiefel) were collected according to the manufacturer’s instructions in horses with recurring non-seasonal urticaria (without the presence and history of clinical signs of IBH) prior to vaccination. Either 6 mm diameter punch biopsies from lesional biopsies (*n* = 6) were collected for paraffin embedding and preparation of hematoxylin-eosin (H&E) stained slides, or 2 mm were collected for RNA isolation and subsequent qPCR. Two mm punch biopsies were collected from lesional (L) (*n* = 12) and non-lesional (NL) (*n* = 11) skin of urticaria-affected horses and healthy (H) skin of non-urticaria-affected horses (*n* = 24) and placed into RNAlater™ stabilization solution (Thermo Fischer, Waltham, MA, USA) for RNA extraction, as described previously [10]. Biopsy processing for qPCR, including the primers used, was described in [11]. The levels of mRNA expression from genes involved in eosinophil recruitment were quantified in both samples of urticaria-affected horses, and compared to healthy skin.

### 2.5. eIL-5-CuMV-TT Vaccine Production and Vaccination of Horses

This was described in [6]. Briefly, eIL-5 homodimers were coupled to VLP CuMV-TT using an SMPH crosslinker, and uncoupled eIL-5 was removed by size exclusion chromatography. Horses were vaccinated subcutaneously using 0.3 mg in 1 mL of eIL-5-CuMV-TT vaccine without the presence of adjuvants. Placebo horses in the IBH trial received 1 mL of PBS. In the first vaccination year, horses received a basic vaccination regimen consisting of three vaccine injections in weeks 0, 4, and 18 ± 2. In subsequent vaccination years, horses received a single yearly booster.

### 2.6. Statistics

All graphs comparing vaccinated horses versus placebo horses show the mean and standard error of the mean (SEM). Statistical analysis by the Kruskal–Wallis test corrected for multiple comparisons. Considered to be statistically significant were *p*-values lower than 0.05: * *p* < 0.05; ** *p* < 0.01; *** *p* < 0.001; **** *p* < 0.0001. 

## 3. Results

### 3.1. Clinical Effect on Seasonal Urticaria in Interleukin (IL)-5 Vaccinated IBH Horses

Commonly, horses with a single hypersensitivity such as IBH, severe equine asthma (formerly known as recurrent airway obstruction (RAO)), or urticaria are predisposed to develop additional allergies and may suffer from multiple hypersensitivities [12]. As such, in a clinical study with horses affected by IBH [6,7,8], we noted that three horses, in addition to IBH, also showed re-occurring seasonal urticaria during summer, which was most probably insect-related. During the course of the long-term study, horses showed urticaria flare-ups in an untreated season and a placebo-treated season (Figure 1A,C). Interestingly, once vaccinated using the eIL-5-CuMV-TT vaccine targeting eosinophilia, horses stopped showing clinical signs of urticaria during all four observed and vaccinated seasons (Figure 1A,C). Along the same lines, a trend towards higher eosinophil levels in blood was found when horses showed more severe clinical signs of urticaria (Figure 1B).

### 3.2. Eosinophilic Gene Expression and Eosinophil Infiltration into Urticaria-Affected Skin

The unexpected finding of preventing re-occurring seasonal urticaria in horses when targeting self-IL-5 using the eIL-5-CuMV-TT vaccine led to the recruitment of horses affected by recurrent urticaria in the absence of clinical signs of IBH. 

At first, skin punch biopsies were collected from healthy horses and non-IBH recurrent urticaria horses showing clinical signs of urticaria at the time point of biopsy collection, with a special interest in levels of mRNA expression from genes involved in eosinophil recruitment. Eosinophilic chemotactic chemokines eotaxin-2 (Figure 2B) and eotaxin-3 (Figure 2C) were significantly upregulated in lesional urticaria-affected skin when compared to healthy skin. Eotaxin-1 (Figure 2A) showed no difference in expression levels among the groups. IL-5 was significantly increased in both urticaria-affected samples when compared to healthy samples (Figure 2D). Comparably, CCR5 (Figure 2E) and CXCL10 (Figure 2F), which were both also involved in the recruitment of eosinophils [11], were found to be significantly increased in both urticaria-affected samples when compared to healthy samples. When comparing the predictive values for these genes, the highest sensitivity and specificity were found for CXCL10, CCR5, eotaxin-2, and IL-5 (Figure 2G, Table 1).

Secondly, skin punch biopsies from six urticaria-affected horses were collected for histological analysis on H&E-stained tissue. Three biopsies of lesional skin showed vast numbers of perivascular eosinophils and lymphoplasmacellular infiltrates in the dermis (Figure 3A,B), and three biopsies of lesional skin were without eosinophil involvement (Figure 3C,D).

### 3.3. Two Case Reports of IL-5 Vaccination in Non-IBH Horses with Non-Seasonal Recurrent Urticaria

Two horses with non-intermittent recurrent urticaria were included in this case series, being vaccinated using the eIL-5-CuMV-TT vaccine. Both horses received two initial prime-boost vaccinations in weeks 0 and 4, with follow-up booster vaccinations to prolong immunity. For both horses, husbandry conditions, feed, and pasture habits remained unchanged during the whole duration of the study. Both horses were regularly vaccinated and dewormed.

Horse 1 (*2011, Fell-pony/Appaloosa mixed breed) was suffering for approximately two years from almost non-intermittently severe recurrent urticaria. This horse did not show any clinical signs of IBH. Urticaria episodes were reoccurring during the whole year, notably during all four seasons of the year, however, the trigger was unknown. With time, urticaria episodes were lasting longer and were getting more severe. Only high doses of corticosteroids were able to cure the symptoms. Nevertheless, due to frequent and severe relapses, the horse was enrolled to participate in our case study to test the eIL-5-CuMV-TT vaccine in horses with recurrent non-seasonal urticaria. Eosinophil levels were enhanced at early and initial time points of urticaria episodes. The horse was vaccinated in weeks 0, 4, and 20. Interestingly, the subsequent day after receiving the booster of the second and third vaccination, the horse showed urticaria wheals around the injection site, which then disappeared the following day. Following the second vaccination onwards using eIL-5-CuMV-TT, eosinophil counts in blood declined, and clinical signs of urticaria disappeared, thus, replacing highly effectively the corticosteroid treatment, and even preventing new episodes of urticaria (Figure 4A). The horse stayed remission-free for two years and then the first urticaria flare-up post-vaccination appeared with severe clinical signs at the head and neck, and eosinophil levels in the blood increased to 0.67 × 10^9^/L (normal range 0.01–0.32 × 10^9^/L). One week later, the horse received an eIL-5-CuMV-TT booster vaccination. Within one week, the de-novo wheal formation was stopped, and existing clinical signs started to heal and disappear quickly. Approximately one year later, another urticaria flare-up appeared and the horse received another booster vaccination three weeks after the flare-up started. This time, when the booster vaccination was applied later during the course of the flare-up, it took much longer, approximately four weeks, until the horse was cleared from urticaria.

Horse 2 (*2003, Oldenburger) was suffering for several years from severe recurrent urticaria. This horse never showed any clinical signs of IBH. Urticaria episodes were reoccurring during the whole year, with a peak in autumn; the trigger was unknown. With time, urticaria episodes were lasting longer and were getting more severe. Only high doses of corticosteroids were able to cure the symptoms for a short duration. A skin punch biopsy was taken by the treating veterinarian, showing a high number of perivascular eosinophils. Due to frequent and severe relapses, the horse was enrolled to participate in our case series to test the eIL-5-CuMV-TT vaccine in horses with recurrent non-seasonal urticaria. The horse was vaccinated in weeks 0 and 4. Following the second vaccination onwards using the IL-5-CuMV-TT, the clinical signs of urticaria disappeared (Figure 4B). A booster vaccination was injected in week 12. In the subsequent year, the horse received a booster vaccination in February, however, presented with a strong flare-up from October until June in the following year. In order to prevent the severe autumn flare-up, the horse was then boosted again in August to achieve a high antibody titer in autumn. Indeed, there was no urticaria afterwards during all of autumn, winter and spring, until today. As such, a booster prior to the expected main disease season was able to prevent new episodes of urticaria.

## 4. Discussion

Our data shed light on an underestimated outsider in the pathogenesis of urticaria in horses. Eosinophils and their recruitment into the skin seem to be closely linked with urticaria lesions. Gene expression profiles highlight eotaxin-2, eotaxin-3, IL-5, CXCL10, and CCR5 and, thus, remind us of similar patterns recently described in horses with seasonal IBH [11]. Interestingly, not all histological skin sections revealed eosinophil infiltration, which was found in 50% of biopsies only. Furthermore, the eIL-5-CuMV-TT vaccine successfully treated and prevented urticaria flare-ups in seasonal, as well as non-intermittent non-seasonal, chronic urticaria, and also in long-term studies with up to four years of vaccination follow-ups. This suggests a leading role of eosinophils in the development and progression of urticaria in horses. The clinical efficacy of vaccine-induced neutralizing anti-IL-5 antibodies, which in turn limit eosinophil numbers in blood and hence limit eosinophil availability for tissue migration [7], may depend on critical timing, and showed enhanced efficacy in the preventive vaccination scenario. A limitation of this study is the small number of patients.

To date, besides the histopathological presence of eosinophils in lesional urticaria biopsies, the role of eosinophils in urticaria in horses is poorly understood. On a general note, depending on the degranulation mechanism, degranulated eosinophils are no longer visible in histological skin sections, which is in contrast to degranulated mast cells [13,14,15]. Hence, eosinophil presence on histological slides may depend on the degranulation status of eosinophils, the time point of collection, and/or may be missed in particular at the end of a flare-up. Bringing the histological findings with a 50% presence of eosinophils into context with the qPCR data showing similar eosinophilic gene expression for lesional, as well as non-lesional, samples during an urticaria flare-up, it may rather point towards a technical issue during preparation causing the degranulation-mediated disappearance of eosinophils from histological slides. As such, judging eosinophil involvement from a histological slide may be difficult. For humans, the role of eosinophils in urticaria is somewhat controversially discussed [16,17]. Approved options to treat humans affected by chronic spontaneous urticaria (CSU) are antihistamines in up to a fourfold dosage and an anti-IgE monoclonal antibody, omalizumab, which have demonstrated efficacy, however, not in all patient groups [16,18]. Even though eosinophil involvement in urticarial wheals is, without doubt, emphasized in various reports, therapies that target the IL-5/eosinophil pathway have not yet shown conclusive results in placebo-controlled randomized trials in patients with CSU [16]. Nevertheless, case studies were able to show successful treatment using anti-IL-5 monoclonal antibodies. In 2018, a human patient with severe refractory eosinophilic asthma and CSU was successfully treated with anti-IL-5 monoclonal mepolizumab [19]. During CSU flares, some studies report blood eosinophilia, while others report eosinopenia. Eosinopenia is not, per se, contradicting eosinophilia, because eosinophils might leave the blood to be recruited to the skin [20]. This indicates that eosinophilia in blood or skin may be very much dependent on the time point when the sample is taken. Along the same lines, our data presented here and the controversial data in humans using anti-IL-5-directed biologicals such as benralizumab or mepolizumab [21,22] may suggest that the timing of therapy application might be a relevant factor for the efficacy of an anti-IL-5/eosinophil targeting. 

Furthermore, potential crosstalk between eosinophils and mast cells and/or basophils might be affected when reducing the number of eosinophils. Our group previously published a significant reduction of basophils in the blood upon eIL-5-CuMV-TT vaccination in horses affected by IBH [23]. The underlying mechanism of such a bystander in basophil reduction is not yet understood. It is possibly explained either by the reduced number of eosinophils, which indirectly affects or diminishes the crosstalk of cells and hence lowers the production of basophils, or by the removal of IL-5 may directly, as basophils express the IL-5Ra [24,25,26,27]. Moreover, a human patient with eosinophilic asthma and an idiopathic mastocytosis receiving mepolozumab showed an eosinophil reduction and surprisingly a parallel significant reduction of mast cell-derived tryptase, strongly linked to mepolizumab applications, thus, further suggesting an eosinophil-mast cell interaction controlled by IL-5 [28].

## 5. Conclusions

In summary, eosinophils may play an unexpected emerging role in urticaria in horses. Blocking IL-5 has shown a very good safety profile both in humans [29,30] and in horses [8]. In particular, the eIL-5-CuMV-TT has been shown to induce reversible antibody titers that require a periodic vaccine booster to maintain antibody titers. To date, equine recurrent urticaria has been challenging for veterinarians and owners, and limited information is available regarding the long-term management of this condition [31]. Hence, exploring new promising treatment options such as the eIL-5-CuMV-TT vaccine targeting eosinophils will be important to treat, prevent and cure recurrent urticaria in horses in the future. In particular, the timing of administration of IL-5-blocking agents might be an important parameter to monitor and investigate in future efficacy studies.

## Figures and Tables

**Figure 1 vaccines-12-00562-f001:**
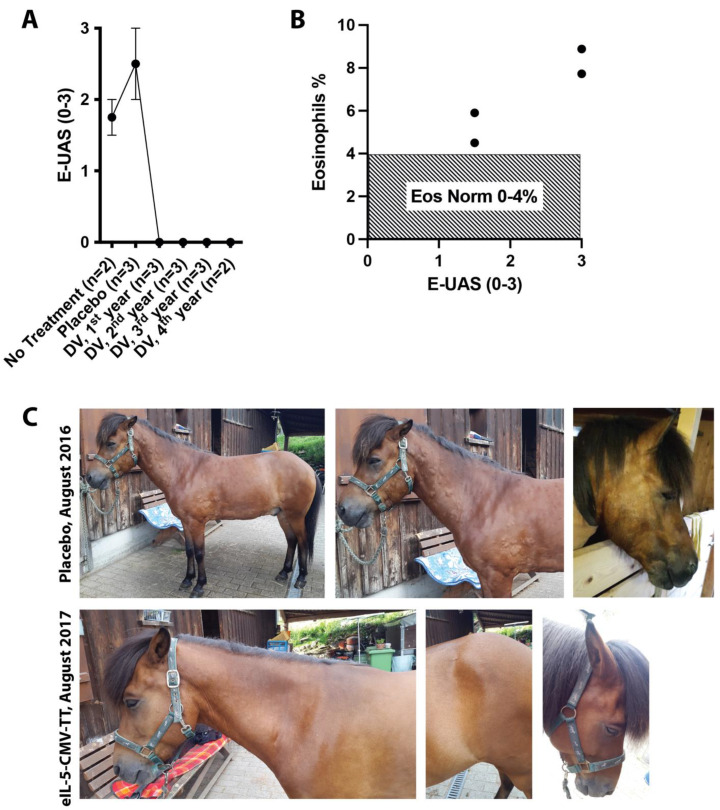
Clinical effect on seasonal urticaria in IL-5 vaccinated IBH horses. Horses affected by insect-bite hypersensitivity (IBH) participated in a vaccination trial using the eIL-5-CuMV-TT vaccine. Three horses in addition showed re-occurring seasonal urticaria. (**A**) E-UAS showing the highest urticaria activity of horses in untreated (no treatment), placebo-treated, or eIL-5-CuMV-TT (DV) in 1st, 2nd, 3rd, and 4th vaccination years. (**B**) E-UAS and contemporaneous eosinophil levels in blood from the three horses in placebo and/or untreated year during urticaria flare-up. (**C**) Representative photographs of a horse during placebo and vaccine treatment at the same time point in both years (placebo, upper line; eIL-5-CuMV-TT vaccinated, lower line).

**Figure 2 vaccines-12-00562-f002:**
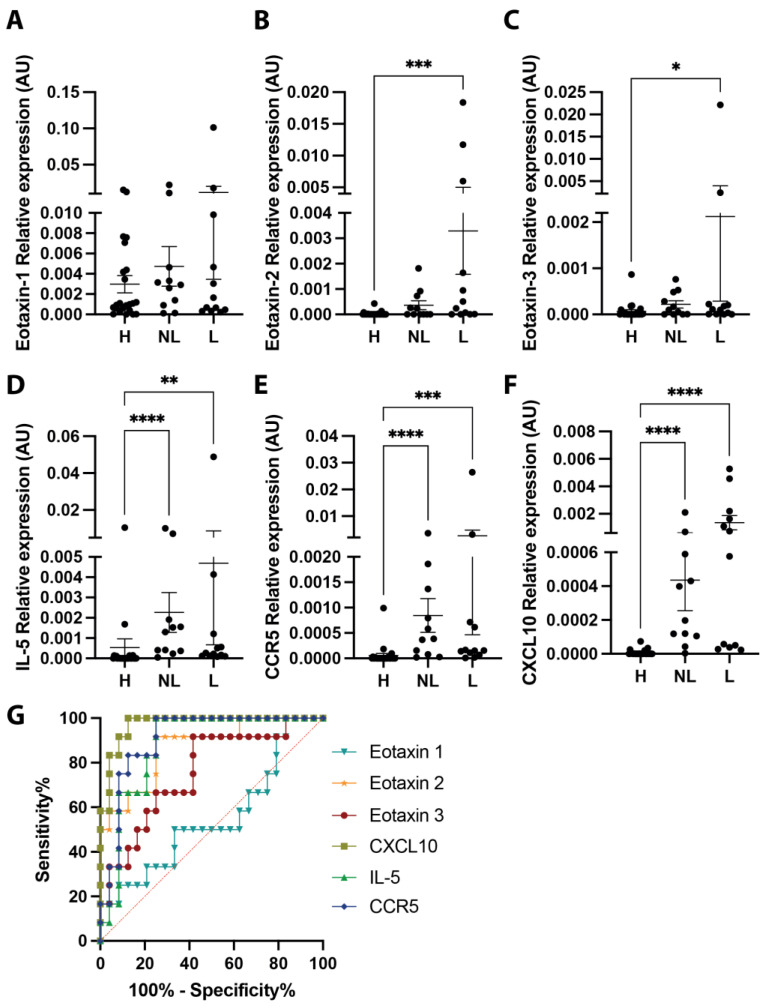
Eosinophilic gene expression in urticaria-affected skin. Relative mRNA expression of eosinophilic genes eotaxin-1 (**A**), eotaxin-2 (**B**), eotaxin-3 (**C**), IL-5 (**D**), CCR5 (**E**), and CXCL10 (**F**) in healthy horses (H, *n* = 24) and urticaria-affected (non-IBH) horses with lesional (L, *n* = 12) and non-lesional (NL, n = 11) biopsies including sensitivity/specificity ROC analysis per gene for lesional versus healthy biopsies (**G**). * *p* < 0.05; ** *p* < 0.01; *** *p* < 0.001; **** *p* < 0.0001.

**Figure 3 vaccines-12-00562-f003:**
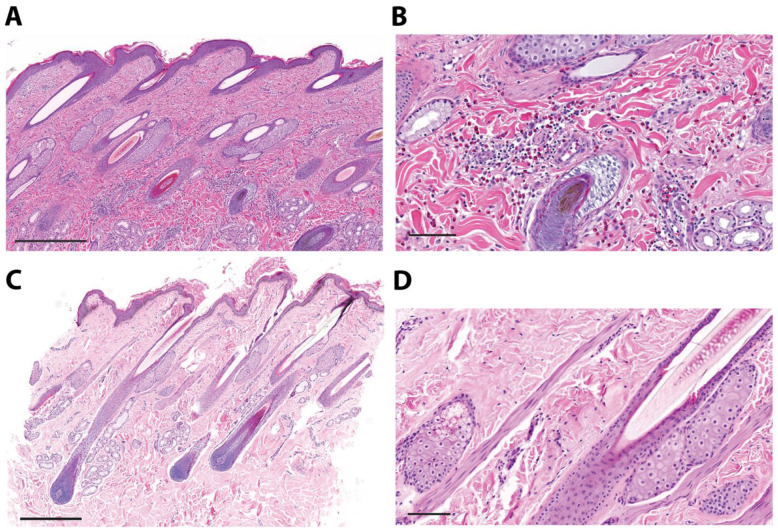
Eosinophil infiltration into urticaria-affected skin. Two representative skin biopsies of a fresh lesion from two urticaria-affected (non-IBH) horses stained with H&E. An overview of the skin biopsy (scale bar = 500 μm) of horse x (**A**) and horse y (**C**) and corresponding enlarged section (scale bar = 100 μm) of horse x (**B**) and horse y (**D**) are shown. (**A**,**B**) shows moderate lymphoplasmacellular and eosinophilic perivascular inflammation, (**C**,**D**) no evidence of inflammatory infiltrates.

**Figure 4 vaccines-12-00562-f004:**
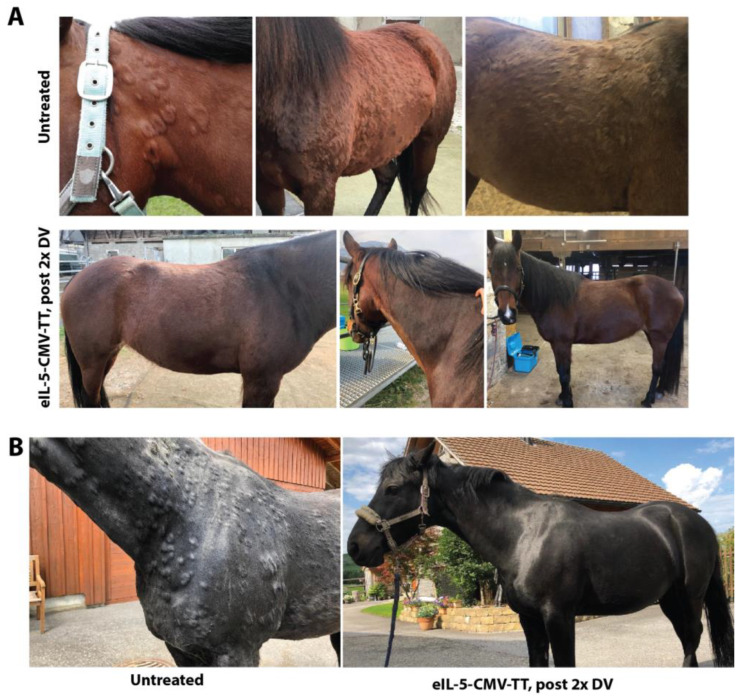
Clinical efficacy of IL-5 vaccination in non-IBH horses with non-seasonal recurrent urticaria. (**A**) Untreated, photographs of urticaria wheels before vaccination; post 2x eIL-5-CuMV-TT vaccinated, representative photographs of horse skin following second vaccination using eIL-5-CuMV-TT. (**B**) Before, photographs of urticaria wheels before vaccination; post 2x eIL-5-CuMV-TT vaccinated, representative photographs of horse skin following second vaccination using eIL-5-CuMV-TT.

**Table 1 vaccines-12-00562-t001:** Sensitivity and specificity ROC analysis per gene for lesional versus healthy biopsies.

Gene/AUC	Area	Standard Error	95% Confidence Interval	*p* Value
Eotaxin 1	0.5486	0.1069	0.3391–0.7581	0.6385
Eotaxin 2	0.8715	0.06312	0.7478–0.9952	0.0003
Eotaxin 3	0.7569	0.08557	0.5892–0.9247	0.013
CXCL10	0.9722	0.02278	0.9276–1	<0.0001
IL-5	0.8785	0.05782	0.7652–0.9918	0.0003
CCR5	0.9063	0.04992	0.8084–1	<0.0001

## Data Availability

Dataset available on request from the authors.

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
