# Peer review of "Eosinophils Play a Surprising Leading Role in Recurrent Urticaria in Horses"

_vaccines, 2024, doi:10.3390/vaccines12060562_

Round 1

Reviewer 1 Report

Comments and Suggestions for Authors

This is a case series of horses suffering from urticaria (+/- insect bite hypersensitivity), who benefit from the eIL-5CMV-TT vaccine. The authors make a very compelling observation since treated animals stop suffering from urticaria. Yet, as a non-expert in the veterinary field, I don’t know how easy it is to distinguish seasonal urticaria lesions from local reactions associated with IBH (e.g, line 48 “we noted that three horses in addition to IBH also showed re-occurring seasonal urticaria during summer, most probably insect-related”.  Maybe some pictures could help. It is also unclear whether the data shown in Figures 2 and 3 are from non-IBH horses with non-seasonal recurrent urticaria, from IBH, or both. What is the eosinophil count in the non-IBH horses? Would it not be more suitable to report all groups since Figure 1? 

Minor comments

It would be nice to have a small introduction about the type of vaccine administered and the previously published results obtained in IBH. This is briefly mentioned in the results section but should be rather included in the introduction (in my opinion). 

I recommend preparing a small flowchart summarizing the overall number of animals with IBH (from the placebo, untreated, and vaccinated groups), those with IBH and urticaria, and those with urticaria only in each group. 

Figure 1A, is it not preferable to show the UAS score for each animal over time? 

In Figure 1B, should the authors not show absolute numbers? Figure A shows 5 animals with positive E-UAS scores, but only 4 are shown in Figure 1B. 

Figure 2, why not showing results from the same animal before and after vaccination? 

Figure 3, since there are no differences in the gene expression in NL and L, it would be important to correlate those results with the histological findings. What means representative skin biopsies? From how many? Which group? With or without IBH? Would counting the eosinophils help? 

Reviewer 2 Report

Comments and Suggestions for Authors

The authors present a study in which a vaccine blocking IL-5 results in unexpected effects on urticaria and eosinophil recruitment to skin lesions associated with urticaria in horses. The materials and methods section should include details on how these horses were vaccinated. Some of the materials and methods are given in the results section. The results section should only include results and must therefore be improved. There is no description of the potential side effects of vaccination in the manuscript. I would like this to be discussed. I am also a bit confused about what was found in this study and what was the results of previous work. Can you please make this a bit clearer in the text?

Line 19: Managing these horses is frustrating – This sentence should be amended. Frustration is a subjective feeling. Please rephrase.

Line 45: The word aseptic, should be added after interstitial

Line 60: Drop frustrating

Materials and methods:               Horses: The geographical study location should be mentioned.

Blood withdrawal: How much blood was collected and what type of anticoagulant/none was used.

The vaccination protocol should be described. How the vaccine was given etc. What was the placebo vaccine?

Line 111: RAO has not previously been described and should be explained.

Line 111-113: Commonly, horses with a single hypersensitivity such as IBH, RAO or urticaria are 111 predisposed to develop additional other allergies and as such often suffer from multiple 112 hypersensitivities (Kehrli et al., 2015). This sentence should be moved to the introduction as it is not a result.

Line 118-120: In the 1st vaccination year horses received a basic vaccination regimen con- 118 sisting of three vaccine injections in week 0, 4 and 16, and in following years horses re- 119 ceived a yearly booster. This should be moved to the materials and methods section.

Line 120: Surprisingly is not recommended to be used in scientific papers.

Line 137-140: At first, skin punch biopsies were collected from healthy horses (H) and horses with 137 clinical signs of urticaria selecting non-lesional (NL) and lesional (L) skin samples. Levels 138 of mRNA expression from genes involved in eosinophil recruitment were quantified in 139 both samples of urticaria horses and compared to healthy skin. This should be moved to the materials and methods section.

Line 165: mis-spelling infiltrates.

Line 167-172: The description of how vaccines were given should be moved to the materials and methods.

Line 273: Surprisingly should be replaced by other word.

In the summary (line 275), a discussion about the possible side effects of this treatment (blocking IL-5) could or should be included.

The study is interesting and communicates a potential treatment for urticaria. 

Comments on the Quality of English Language

The english is fairly good, but some rephrasing is recommended in general. 

Reviewer 3 Report

Comments and Suggestions for Authors

This is a novel and important report regarding potential protection of horses against the serious condition of recurrent urticaria, independent of cause.  This study used a previously reported vaccine to determine efficacy of treatment in horses with clinical evidence of recurrent urticaria.  The report is reasonably well prepared, details are somewhat limited, but the reported data is compelling to support consideration for use in equine patients.  In general, there is substantial editing that is needed to improve the clarity of writing in this report. 

Specific comments:

Lines 54-55, appropriate use of parentheses should be adopted, there are extras (bacterial and perdum), protozoal

Line 59, the authors need to provide complete sentences.  As written, this is not a complete sentence.  Suggestion, A challenge for the long-term clinical management and cure of equine recurrent urticaria is identification of the underlying cause. 

Line 60, Identification of the underlying cause and etiology of equine recurrent urticaria can be challenging because recurrence is common.

Line 62-63, additionally, chronic steroid administration is associated with immune suppression.  The authors should include this risk of steroid administration.

Line 74 M&M, there is little detail regarding study horse population, number, type or any detail.  It is impossible to determine sample size for any part of this investigation. 

Line 87, this should be listed as Blood collection rather than withdrawal.

Line 94, there should be some description of how biopsy samples were collected.    

Line 102, there needs to be some description of the vaccine.  Briefly, as reported then a description of the vaccine. 

Line 111, current nomenclature is Severe Equine Asthma rather than RAO. 

Line 112, to develop additional allergies and may suffer from multiple hypersensitivities

Line 234, To date

Line 246, wheals is without doubt emphasized  

Line 250, monoclonal antibodies

Line 274, To date, equine recurrent urticaria has been challenging for veterinarians and owners, limited information is available regarding long term management for this condition.   

Comments on the Quality of English Language

The quality of writing is fair, there is substantial editing required to improve the clarity of writing.  Much of this editing likely relates to use of English in this report. 

Round 2

Reviewer 1 Report

Comments and Suggestions for Authors

The manuscript is now more clear, and the authors adequately addressed the concerns.